# A SIMPLE AND INTERPRETABLE MODEL OF GROKKING MODULAR ARITHMETIC TASKS

## ABSTRACT

We present a simple neural network that can generalize on various modular arithmetic tasks such as modular addition or multiplication, and exhibits a sudden jump in generalization known as *grokking*. Concretely, we present (i) fully-connected two-layer networks that exhibit grokking on various modular arithmetic tasks under vanilla gradient descent with the MSE loss function in the absence of any regularization; (ii) evidence that grokking modular arithmetic corresponds to learning specific representations whose structure is determined by the task; (iii) *analytic* expressions for the weights – and thus for the embedding – that solve a large class of modular arithmetic tasks; and (iv) evidence that these representations are also found by gradient descent as well as AdamW, establishing complete ("mechanistic") interpretability of the representations learnt by the network.

## 1 INTRODUCTION AND OVERVIEW OF LITERATURE

Grokking is an effect discovered empirically in Power et al. (2022). Its phenomenology is characterized by a steep and delayed rise in generalization from $0\%$ to a fixed value, as depicted in Fig. 1b. Beyond that observation, however, there are no clear characteristics of grokking that are reproduced across different works. We start with a lightening review of various claims made in the literature.

In the original work Power et al. (2022), the authors studied how a shallow transformer learns data distributions that are generated by simple deterministic rules (termed 'algorithmic datasets'). Examples of such datasets include modular arithmetic, finite groups, bit operations and more. Specifically, in Power et al. (2022) the data took a form of a string "$a \circ b = c$", where $c$ was masked and had to be predicted by a two-layer decoder-only transformer. In that study, the following empirical facts were observed:

- Generalization occurs long after training accuracy reached $100\%$. The jump in generalization is quite rapid and occurs after a large number of epochs (cf. Fig. 1).
- There is a minimal amount of data (dependent on the task) that needs to be included into the training set in order for generalization to occur (cf. Fig. 4b).
- Various forms of regularization improve how quickly grokking happens. Weight decay included in AdamW optimizer showed to be particularly effective (cf. Fig. 4b).

In subsequent work Liu et al. (2022a), the authors simplified the architecture to a single linear learnable encoder followed by a multilayer perceptron (MLP) decoder and showed that, even if the task is recast as a classification problem, grokking persists. They also interpret grokking as a competition between encoder and decoder, and developed a toy model of grokking as dynamics of the embeddings only. This model indeed leads to some quantitative predictions such as the critical amount of data needed for grokking to happen relatively fast.

In more recent work Thilak et al. (2022), it was argued that if the Adam optimizer is used, then in order for grokking to happen without regularization, the training dynamics must undergo a slingshot – a sudden explosion in the training loss – which was followed by the rise of generalization. It was further shown that those slingshots and grokking can be turned on and off by tuning the $\epsilon$ parameter of the Adam optimizer.

In a blogpost Nanda and Lieberum (2022), it was argued that the algorithm for modular addition learnt by a single-layer transformer can be reverse-engineered and is human-interpretable. It was

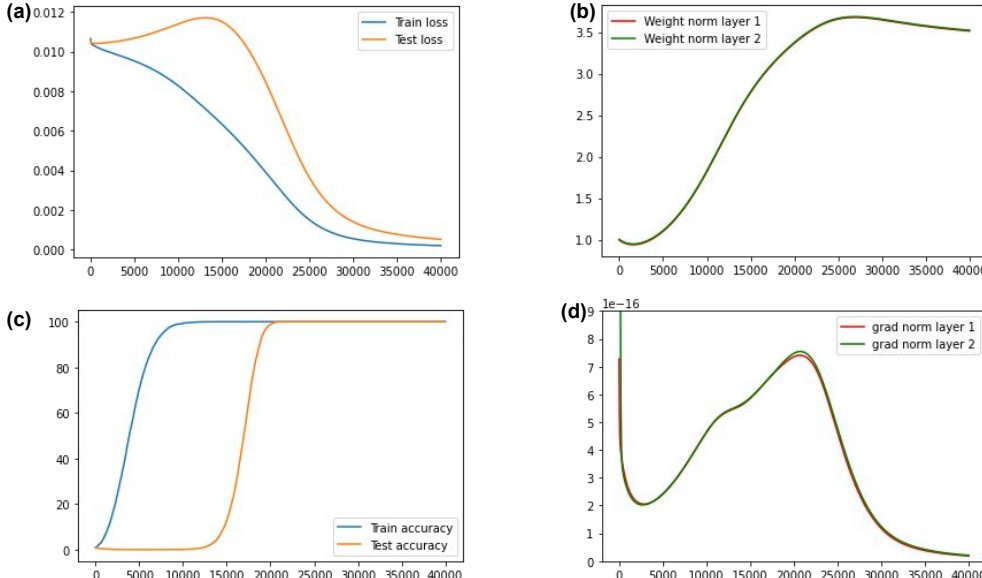

Figure 1: Dynamics under GD for the minimal model equation 4 with MSE loss and $\alpha = 0.49$. **(a)** Train and test loss. Train loss generally decays monotonically, while test loss reaches its maximum right before the onset of grokking. **(b)** Norms of weight matrices during training. We do not observ a large increase in weight norms as in Thilak et al. (2022), but we do see that weight norms start growing at the onset of grokking. **(c)** Train and test accuracy showing the delayed and sudden onset of generalization. **(d)** Norms of gradient vectors. The dynamics accelerates until the test loss maximum is reached and then slowly decelerates.

further argued that (i) regularization is required for grokking and (ii) there should be no grokking in the infinite-data regime. Furthermore, many other algorithmic datasets were considered [1].

On a theoretical front, the authors of Barak et al. (2022) studied *online* learning of the $(k, n)$ sparse parity problem where the network function is asked to compute parity of $k$ bits in a length-$n$ string of random bits. In particular, they observed grokking both in under- and over-parametrized regimes. For large minibatch sizes, generalization was attributed to amplification of the information already present in the initial gradient (called Fourier gap) rather than to the diffusive search by stochastic gradient descent, and derived the scaling of grokking time with $n, k$ to be $n^{O(k)}$.

Finally, Liu et al. (2022b) studied grokking for non-algorithmic datasets and its dependence on the initialization, while Žunkovič and Ilievski (2022) developed a solvable model of grokking in the teacher-student setup. Similar training dynamics was observed in Arous et al. (2021).

To summarize, the available results, although undoubtedly inspiring, leave grokking on algorithmic datasets as a somewhat mysterious effect. Furthermore, the empirical results suggest that grokking provides a fascinating platform for quantitatively studying many fundamental questions of deep learning in a controlled setting. These include: (i) the precise role of regularization in deep nonlinear neural networks; (ii) feature learning; (iii) the role of training data distributions in optimization dynamics and generalization performance of the network; (iv) data-, parameter- and compute-efficiency of training; (v) interpretability of learnt features; and (vi) expressivity of architectures and complexity of tasks.

This motivates the present study, proposing and analyzing a minimal yet realistic model and optimization process that lead to grokking on modular arithmetic tasks.

---

[1]From our perspective the insightful work Nanda and Lieberum (2022) is viewed as empirical: periodic structure in the activations was discovered experimentally and then reasoned about qualitatively. In contrast, the present work is theoretical: We describe an analytically *solvable* model and then show experimentally that our analytic solution captures main properties of the experiment. Nanda *et. al.* expanded the blogpost into Nanda et al. (2023) after the present article appeared.

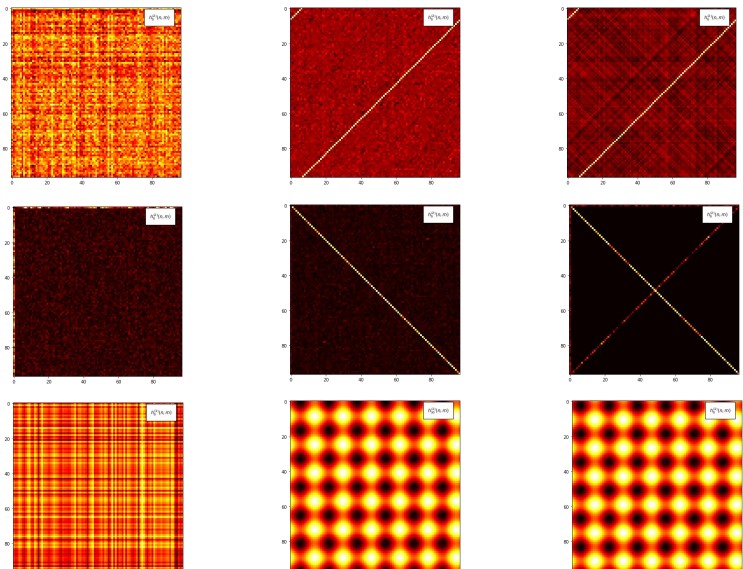

Figure 2: Preactivations. **First row**: Preactivation $h_6^{(2)}(n,m)$. **Second row**: Fourier image of the Preactivation $h_6^{(2)}(n,m)$. **Third row**: Preactivation $h_6^{(1)}(n,m)$ or $h_{30}^{(1)}(n,m)$. **First column**: At initialization. **Second column**: Found by vanilla GD. The Fourier image shows a single series of peaks corresponding to $m + n = 6$ mod 97. **Third column**: Evaluated using the analytic solution equation 6-equation 7. The Fourier image shows the same peak as found by GD, but also weak peaks corresponding to $2m = 6$ mod 97, $2n = 6$ mod 97 and $m - n = 6$ mod 97 that were supressed by the choice of phases via equation 12.

## 2   SET UP AND OVERVIEW OF RESULTS

In this Section we describe a very simple, solvable and interpretable, setting where grokking takes place and learnt features can be understood analytically. We consider a two-layer MLP network without biases, given by

$$h_k^{(1)}(x) = \sum_{j=1}^{D} W_{kj}^{(1)} x_j \,, \qquad z_i^{(1)}(x) = \phi(h_i^{(1)}(x)) \,, \tag{1}$$

$$h_q^{(2)}(x) = \frac{2}{N} \sum_{k=1}^{N} W_{qk}^{(2)} z_k^{(1)}(x) \,, \tag{2}$$

where $N$ is the width of the hidden layer, $D$ is the input dimension, and $\phi$ is an activation function. At initialization the weights are sampled from the standard normal distribution $W^{(1)}, W^{(2)} \sim \mathcal{N}(0, 1)$. In equation 1–equation 2, we have chosen to follow the mean-field parametrization Song et al. (2018): this parametrization ensures that the analytic solution presented in the next Section remains finite in the large-$N$ limit[2].

Given this architecture, we then set up modular arithmetic tasks as classification problems. To this end, we fix an integer $p$ (that does not have to be prime) and consider functions over $\mathbb{Z}_p$. Each input integer is encoded as a one-hot vector. The output integer is also encoded as a one-hot vector. For the task of learning bivariate functions over $\mathbb{Z}_p$ the input dimension is $2p$, the output dimension is $p$, the total number of points in the dataset is $p^2$, while the model equation 1–equation 2 has $3Np$ parameters. Finally, we split the dataset $\mathcal{D}$ into train $\mathcal{D}_{\text{train}}$ and test $\mathcal{D}_{\text{test}}$ subsets, and furnish this setup with the MSE loss function.[3]

---

[2]In the limit of infinite width, the meanfield parametrization allows for feature learning. However, here we only consider large $N$ limit of the network *in the end of training*.

[3]CSE loss can be used, if desired.

Under this minimal setting grokking occurs consistently for many modular functions, provided enough epochs of training have taken place *and* the fraction of data used for training,

$$\alpha \equiv \frac{|\mathcal{D}_{\text{train}}|}{|\mathcal{D}|}\,, \tag{3}$$

is sufficiently large (if $\alpha$ is too small, generalization is not possible even after long training time). By adjusting width $N$, at fixed $\alpha$, we can tune between underparametrized and overparametrized regimes. The 'simplest' optimizer that leads to grokking is the full-batch gradient descent. *No explicit regularization is necessary for grokking to occur.* We have tried other optimizers and regularization methods such as AdamW, GD with weight decay and momentum, SGD with Batchnorm, and GD with Dropout. Generally, regularization and the use of adaptive optimizers produce two effects: (i) grokking happens after a smaller number of epochs and (ii) grokking happens at smaller $\alpha$. See Fig. 4.

In the case of *quadratic* activation the full network function takes an even simpler form

$$f(x) = \frac{2}{N} W^{(2)} \left( W^{(1)} x \right)^2 \,. \tag{4}$$

This function is *cubic* in parameters and *quadratic* in its inputs. Eq. equation 4 is the simplest possible *nonlinear* generalization of the '$u$-$v$' model studied in Lewkowycz et al. (2020). The exact results are derived for this particular choice (and can be generalized to other monomials if wished) while empirical results are only mildly sensitive to the choice of activation function.

Whether grokking happens or not depends on the modular function at hand assuming the architecture and optimizer are fixed. We show that for any function of the form $f(n, m) = f_1(n) + f_2(m) \mod p$ as well as $\tilde{f}(n, m) = F(f_1(n) + f_2(m)) \mod p$ there is an analytic solution for the weights that yield $100\%$ accuracy and these weights are approximately found by various optimizers with and without regularization. Functions of the form $g(n, m) = g_1(n) \cdot g_2(m) \mod p$ can also be grokked, however the analytic solution is more complex and involves logarithms over $\mathbb{Z}_p$. Functions of the form $f(n, m) + g(n, m) \mod p$ are more difficult to grok: they require more epochs and larger $\alpha$.

In summary, our setup is simple enough to be analytically solvable and, therefore, fully ("mechanistically") interpretable, but complex enough to exhibit representation learning and, consequently, grokking.

## 3 INTERPRETABILITY: ANALYTIC EXPRESSION FOR THE WEIGHTS

### 3.1 MODULAR ADDITION

In this Section we will exhibit the analytic expression for the weights that solve the modular addition task. Namely, the network supplied with these weights implements the following modular function

$$f(n, m) = n + m \mod p\,. \tag{5}$$

This solution is approximate and can be made increasingly more accurate (meaning the test *loss* can be made arbitrarily close to $0$) by increasing the width $N$. To simplify the presentation, we will discuss modular addition at length and then generalize the solution to a broad class of modular functions. In the next Section we will provide evidence that the GD and AdamW find the same solution.

**Claim I.** If the network function has the form equation 4 then the weights $W_{kn}^{(1)}$ and $W_{qk}^{(2)}$ solving the modular addition problem are given by

$$W_{kn}^{(1)} = \begin{pmatrix} \cos\left(2\pi\frac{k}{p}n_1 + \varphi_k^{(1)}\right) \\ \cos\left(2\pi\frac{k}{p}n_2 + \varphi_k^{(2)}\right) \end{pmatrix}^T\,, \qquad n = (n_1, n_2) \tag{6}$$

$$W_{qk}^{(2)} = \cos\left(-2\pi\frac{k}{p}q - \varphi_k^{(3)}\right)\,, \tag{7}$$

where we represent $W_{kn}^{(1)}$ as a row of two $N \times p$ matrices and index $n$ takes $2p$ values as $n_1 = 0, 1, \ldots, p-1$ while $n_2 = p, \ldots, 2p-1$. The full size of $W_{kn}^{(1)}$ is $N \times 2p$. The phases $\varphi_k^{(1)}, \varphi_k^{(2)}$ and $\varphi_k^{(3)}$ are random, sampled from a uniform distribution and satisfy the constraint equation 12.

**Reasoning.** Here we explain why and how the solution equation 6-equation 7 works. There are two important ingredients in equation 6-equation 7. The first ingredient is the periodicity of weights with respect to the indices $n_1, n_2, q$. The set of frequencies is determined by the base of $\mathbb{Z}_p$. The full set of independent frequencies is obtained by varying $k$ from 0 to $\frac{p-1}{2}$ if $p$ is odd and to $\frac{p}{2}$ if $p$ is even. The second ingredient is the set of phases $\varphi_k^{(1)}, \varphi_k^{(2)}, \varphi_k^{(3)}$. Indeed, Eqs. equation 6-equation 7 solve modular addition *only* after these phases are chosen appropriately. We will discuss the choice shortly.

To show that equation 6-equation 7 solve modular addition we will perform the inference step analytically. Consider a general input $(n, m)$ represented as a pair of one-hot vectors stacked into a single vector of size $2p \times 1$.

The preactivations in the first layer are given by (we drop the normalization factors)

$$h_k^{(1)}(n, m) = \cos\left(2\pi\frac{k}{p}n + \varphi_k^{(1)}\right) + \cos\left(2\pi\frac{k}{p}m + \varphi_k^{(2)}\right) . \tag{8}$$

The activations in the first layer are given by

$$z_k^{(1)}(n, m) = \left(\cos\left(2\pi\frac{k}{p}n + \varphi_k^{(1)}\right) + \cos\left(2\pi\frac{k}{p}m + \varphi_k^{(2)}\right)\right)^2 , \tag{9}$$

which, after some trigonometry, becomes

$$
\begin{aligned}
z_k^{(1)}(n, m) &= 1 + \frac{1}{2}\left(\cos\left(2\pi\frac{k}{p}2n + 2\varphi_k^{(1)}\right) + \cos\left(2\pi\frac{k}{p}2m + 2\varphi_k^{(2)}\right)\right) \\
&+ \cos\left(2\pi\frac{k}{p}(n+m) + \varphi_k^{(1)} + \varphi_k^{(2)}\right) + \cos\left(2\pi\frac{k}{p}(n-m) + \varphi_k^{(1)} - \varphi_k^{(2)}\right) (10)
\end{aligned}
$$

Finally, the preactivations in the second layer take form

$$
\begin{aligned}
h_q^{(2)}(n, m) &= \frac{1}{4}\sum_{k=1}^{N}\cos\left(2\pi\frac{k}{p}(2n-q) + 2\varphi_k^{(1)} - \varphi_k^{(3)}\right) + \cos\left(2\pi\frac{k}{p}(2n+q) + 2\varphi_k^{(1)} + \varphi_k^{(3)}\right) \\
&+ \frac{1}{4}\sum_{k=1}^{N}\cos\left(2\pi\frac{k}{p}(2m-q) + 2\varphi_k^{(1)} - \varphi_k^{(3)}\right) + \cos\left(2\pi\frac{k}{p}(2m+q) + 2\varphi_k^{(1)} + \varphi_k^{(3)}\right) \\
&+ \frac{1}{2}\sum_{k=1}^{N}\cos\left(2\pi\frac{k}{p}(n+m-q) + \varphi_k^{(1)} + \varphi_k^{(2)} - \varphi_k^{(3)}\right) \\
&+ \frac{1}{2}\sum_{k=1}^{N}\cos\left(2\pi\frac{k}{p}(n+m+q) + \varphi_k^{(1)} + \varphi_k^{(2)} + \varphi_k^{(3)}\right) \\
&+ \frac{1}{2}\sum_{k=1}^{N}\cos\left(2\pi\frac{k}{p}(n-m-q) + \varphi_k^{(1)} - \varphi_k^{(2)} - \varphi_k^{(3)}\right) \\
&+ \frac{1}{2}\sum_{k=1}^{N}\cos\left(2\pi\frac{k}{p}(n-m+q) + \varphi_k^{(1)} - \varphi_k^{(2)} + \varphi_k^{(3)}\right) \\
&+ \sum_{k=1}^{N}\cos\left(2\pi\frac{k}{p}q + \varphi_k^{(3)}\right) . 
\end{aligned}
\tag{11}
$$

Expression equation 11 does not yet perform modular addition. Observe that each term in equation 11 is a sum of waves with different phases, but systematically ordered frequencies. We are going to choose the phases $\varphi_k^{(1)}, \varphi_k^{(2)}, \varphi_k^{(3)}$ to ensure constructive interference in the third line of equation 11. The simplest choice is to take

$$\varphi_k^{(1)} + \varphi_k^{(2)} = \varphi_k^{(3)} . \tag{12}$$

Then the term in the third line of equation 11 takes form

$$\frac{1}{2}\sum_{k=1}^{N}\cos\left(2\pi\frac{k}{p}(n+m-q)\right) = \frac{N}{2}\delta(n+m-q), \tag{13}$$

where $\delta(n+m-q)$ is the modular version of the $\delta$-function. It is equal to 1 when $n+m-q = 0 \bmod p$ and is equal to 0 otherwise. This concludes the constructive part of the interference.

Next, we need to ensure that all other waves (*i.e.* all terms, but the third term in equation 11) interfere destructively. Fortunately, this can be accomplished by observing that the constraint equation 12 leaves some phases in every single term in equation 11 apart from the third one. We will spare the reader the explicit expression. Every remaining term takes form

$$\frac{1}{2} \sum_{k=1}^{N} \cos \left( 2\pi \frac{k}{p} s + \varphi_k \right) , \tag{14}$$

where $s$ is an integer and $\varphi_k$ is a linear combination of $\varphi_k^{(1)}$ and $\varphi_k^{(2)}$. We now assume that $\varphi_k^{(1)}$ and $\varphi_k^{(2)}$ are uniformly distributed random numbers. Then so are $\varphi_k$. For any appreciable $N$ (see Fig. 4b) we have

$$\sum_{k=1}^{N} \cos \left( 2\pi \frac{k}{p} s + \varphi_k \right) \ll N , \tag{15}$$

which implies that every term in equation 11 can be neglected compared to the third term. Thus, for reasonable values of $N$ (and restoring normalisation) the network function $h_q^{(2)}(n, m)$ takes form

$$h_q^{(2)}(n, m) \approx \frac{1}{2} \sum_{k=1}^{N} \cos \left( 2\pi \frac{k}{p} (n + m - q) \right) = \delta(n + m - q) . \tag{16}$$

In the limit of large $N$ the approximation becomes increasingly more accurate. Note that $h_q^{(2)}(n, m)$ is finite in the infinite width limit.

The test accuracy of the solution equation 6-equation 7 *increases with width*. For larger $N$ the interference is stronger leading to the better approximation of the $\delta$-function and, ultimately, to better accuracy.

We emphasize a curious point: for modular arithmetic tasks larger width *does not* imply a larger set of relevant features. Instead, large width leads to *redundant representations*: Each frequency appears several times with different random phases ultimately leading to a better wave interference. The same effect is observed in the solutions found by an optimizer.

We further emphasize that the weights equation 6-equation 7 are not iid. At fixed $k$ the weights $W_{kn}^{(1)}, W_{qk}^{(2)}$ are *strongly correlated* with each other. This provides a non-trivial yet analytically solvable, interpretable example of a correlated, non-linear, network *far away from the Gaussian limit*.

The weights equation 6-equation 7 also work for other activation functions, including ReLU, however the $100\%$ accuracy is achieved at higher width compared to quadratic activation function (more details in Appendix B). Indicating that the solution we found is very robust.

## 3.2 General modular functions and complexity

The solution equation 6-equation 7 can be easily generalized to represent a general modular function of the form

$$f(n, m) = f_1(n) + f_2(m) \bmod p , \tag{17}$$

where $f_1, f_2$ are arbitrary modular functions of a single variable. The generalization becomes obvious once we observe that the proof presented in Section 3 holds verbatim upon replacing $n \to f_1(n)$ and $m \to f_2(m)$ leading to a $\delta$-function supported on $f_1(n) + f_2(m) - q = 0 \bmod p$. These solutions are also found by the optimizer just like in the case of modular addition. More precisely, we claim

**Claim II.** If the network function has the form equation 4 then the weights $W_{kn}^{(1)}$ and $W_{qk}^{(2)}$ solving the modular arithmetic task $f(n, m) = f_1(n) + f_2(m) \bmod p$ are given by

$$W_{kn}^{(1)} = \begin{pmatrix} \cos \left( 2\pi \frac{k}{p} f_1(n_1) + \varphi_k^{(1)} \right) \\ \cos \left( 2\pi \frac{k}{p} f_2(n_2) + \varphi_k^{(2)} \right) \end{pmatrix} , \qquad n = (n_1, n_2) \tag{18}$$

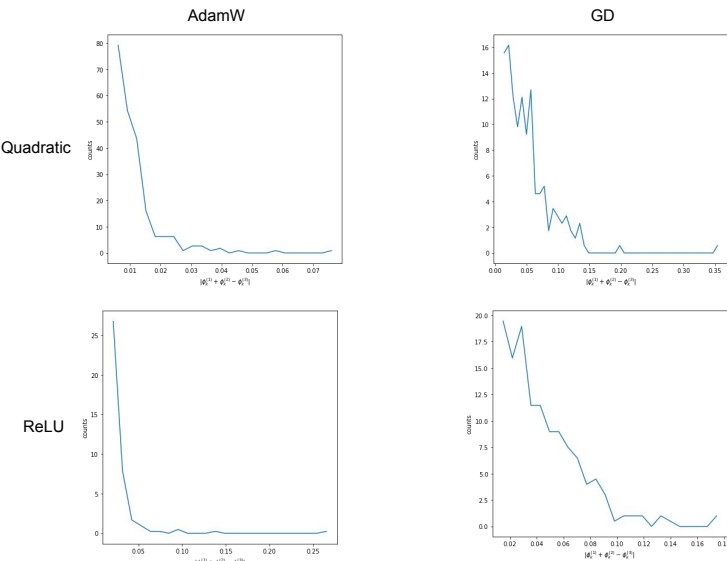

Figure 3: Solutions found by the optimizer. In all cases distribution of $\varphi_k^{(1)} + \varphi_k^{(2)} - \varphi_k^{(3)}$ is strongly peaked around $0$. The solutions found by AdamW are closer to the analytic ones because the phases are peaked stronger around $0$. Note that for solutions found by the optimizer the phases are not iid which leads to the better accuracy.

and equation 7. The weights, again, depend on the modular arithmetic task at hand. Furthermore, for this class of tasks the weights in the readout layer are unchanged. A simple example is the sum of squares $f(n, m) = n^2 + m^2 \mod p$ as well as arbitrary linear function $f(n, m) = An + Bm + C \mod p$. The activations for the sum of squares are presented in the Appendix C.

**Corollary.** Given the Claim II, a more general modular task $\tilde{f}(n, m) = F(f_1(n) + f_2(m)) \mod p$, can be analytically solved, assuming that $F$ is invertible. This is accomplished by modifying the readout layer weights as follows

$$W_{qk}^{(2)} = \cos\left(-2\pi \frac{k}{p} F^{-1}(q) - \varphi_k^{(3)}\right) . \tag{19}$$

This solution approximates $\delta(f_1(n) + f_2(m) - F^{-1}(q))$, which is equivalent to the $\delta$-function supported on the claimed modular task $\delta(F(f_1(n) + f_2(m)) - q)$ assuming $F^{-1}$ is single-valued. Note that application of $F^{-1}$ must follow modular arithmetic rules. If $F^{-1}$ is not single-value then the accuracy will be approximately $100\%/b$, where $b$ is the number of branches. A simple example is $f(n, m) = (n + m)^2$. The activations for this task are presented in the Appendix. Analytic solution has accuracy $\approx 50\%$ since $F^{-1}(x) = x^{\frac{1}{2}} \mod p$, which has two branches.

## 4 PROPERTIES OF SOLUTIONS FOUND BY GRADIENT DESCENT

### 4.1 GENERAL PROPERTIES

In this Section we show that optimization of the network equation 1-equation 2 yields a solution that is very close to the one we proposed in the previous Section.

As can be seen in Fig. 1 during the optimization the network first overfits the train data. The periodic structure in weights and activations does not form at that point. Train loss slowly gets smaller until it either (i) saturates leading to a memorizing solution without grokking the problem, or (ii) after a period of slow decrease, it slightly accelerates. It is during that time grokking and feature formation take place. The test loss is *non-monotonic* and reaches a local maximum right before grokking happens. In the memorizing phase test loss never leaves this local maximum. This general behaviour

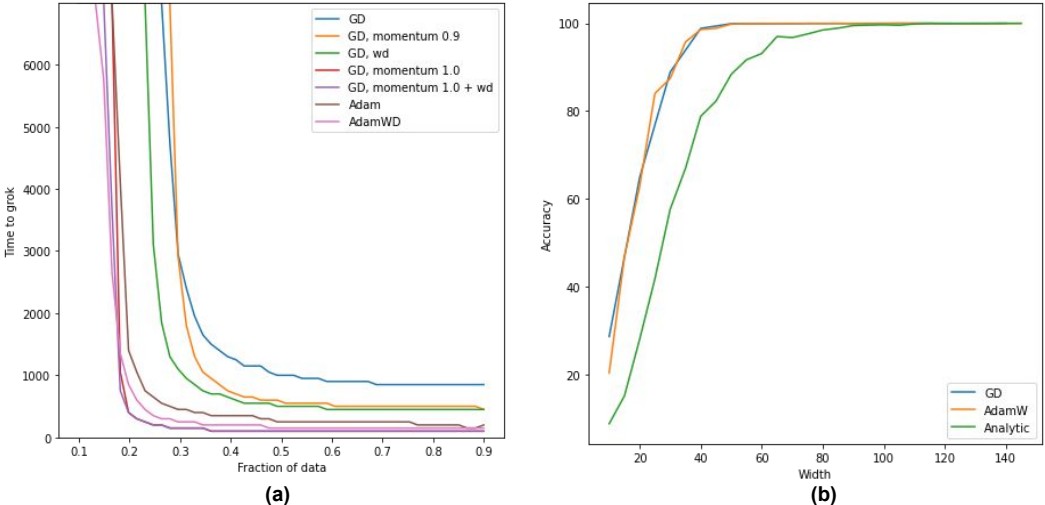

Figure 4: Scaling with width and data. **(a)** Grokking time vs. the amount of training data for various optimizers. The abrupt change in grokking time is observed at different $\alpha$. Momentum appears to play a major role both in reducing grokking time and $\alpha$. **(b)**: Test accuracy as a function of width for the solution found by GD, AdamW and for the analytic solution equation 6–equation 7. The optimizer can tune phases better than random uniform distribution in order to ensure better cancellations. The shape of the curves also depends on the amount of data used for training and number of epochs. Here we took $\alpha = 0.5$ and trained longer for GD.

appears to be insensitive to either optimizer used, loss function or modular function (*i.e.* dataset) in question.

We then show empirically that independently of the optimizer and the loss function the features found by optimization in the grokking phase are indeed periodic functions with frequencies $\frac{2\pi k}{p}$ where $k = 0, \ldots, p-1$. If the width is larger than $\frac{p-1}{2}$ then multiple copies of these functions are found with different phases. The phases are approximately random and satisfy the constraint equation 12 approximately as we show in Fig.3. Given the simplicity of our setup, the basic explanation for grokking must be quite banal: At some point in training (and assuming there is enough data), the only way to decrease the training loss is to start learning the "right" features.

## 4.2 SCALING

Scaling with width and dataset size are presented on Fig. 4. The accuracy of solution equation 6-equation 7 favorably scales with width. This stems from the simple fact that destructive interference condition equation 15 becomes increasingly more accurate with larger $N$. The test accuracy of trained network also increases with the width, reaching perfect accuracy before the analytic solution does, which is not surprising because optimizer can tune the individual phases to ensure better performance.

The grokking time scales with the amount of data. Both for GD and AdamW there is a critical amount of data $\alpha_c$ such that grokking is possible. The precise value of $\alpha_c$ is hard to determine because of the long time scales needed for grokking close to $\alpha_c$. This is clearly seen on Fig.4. AdamW appears to be more data-efficient than GD, however it is difficult to rule out the possibility that for $\alpha \approx 0.2$ GD requires extremely long time scales to show grokking.

# 5  CONCLUSIONS AND DISCUSSIONS

## 5.1  CONCLUSIONS

We have presented a simple architecture that exhibits grokking on a variety of modular arithmetic problems. The architecture equation 4 is simple enough to determine the weights and features that solve modular addition problems analytically, leading to complete interpretability of what was learnt by the model: the network is learning a $\delta$-function represented by a complete set of trigonometric functions with frequencies determined by the base of modular addition; the phases are chosen to ensure that waves concentrated on $m + n = q$ mod $p$ interfere constructively.

Grokking is intimately connected to feature learning. In particular, random feature models such as infinitely-wide neural networks (in the NTK regime) do not exhibit grokking, at least on the tasks that involve modular functions. In addition, Ref. Liu et al. (2022a) argued that grokking is due to the competition between encoder and decoder. While it is certainly true in their model, in the present case there is no learnable encoder, but grokking is still present. In our minimal setup, the simplest explanation for grokking is that once training loss reached a certain value, the only way to further minimize it is to start learning the right features (provided there is enough data).

## 5.2  DISCUSSIONS AND LIMITATIONS

We close with a discussion of open problems and directions.

Different modular functions clearly fit into different complexity classes: (i) functions that can be learnt easily; (ii) functions that can be learnt with a lot of data and training time; and (iii) functions that cannot be learnt at all (at least within the class of architectures we and Power et al. (2022) have considered). It would be interesting to (1) define the notion of complexity rigorously as a computabe quantity and (2) construct architectures/optimizers that can learn more complex modular functions (or argue that it cannot be done).

A neural network can learn a smooth approximation to complicated modular operations, such as modular square root and modular logarithm. It would be interesting to determine if these approximations provide any practical gain over known algorithms that perform these operations as well as to enable the networks to operate over large numbers.

The critical amount of data needed for generalization, $\alpha_c$, is likely to be computable as well, and is a measure of complexity of a modular function. We would like to have an expression for the absolute minimal value of $\alpha_c$ (*i.e.* minimized over all possible ML methods). This value is also an implicit function of modulus $p$, and the modular functions with larger modulus are likely simpler since we find empirically that $\alpha_c$ is a decreasing function of $p$. The value of $\alpha_c$ further depends on how training set is sampled from the entire dataset; the appropriate choice of the sampling method may thus improve the data efficiency.

Presented solutions only hold for the two-layer MLPs. To quantify the role of depth, we would like to have examples of algorithmic tasks that require a deeper architecture. For instance, it is possible that deep convolutional architectures, given an appropriate algorithmic dataset with hierarchical structure, would admit a solution in terms of wavelets  Cheng and Ménard (2021).

In real-world datasets and tasks that require feature learning, it is possible that grokking is happening but the jumps in generalization after learning a new feature may be so small that we perceive a continuous learning curve. To elucidate this point further, it is important to construct a realistic model of datasets and tasks with controllable amount of hierarchical structure. More broadly, it would be very interesting to characterize grokking in terms that are not specific to a particular problem or a particular model and to establish whether it occurs in more traditional ML settings.

Given the simplicity of our model, equation 4, loss function (MSE) and optimization algorithm (vanilla GD), it is plausible that some aspects of the training dynamics – not just the solution at the end of training – can be treated analytically. As the training and test losses show peculiar dynamics, it would be interesting to understand the structure of the loss landscape to explain the dynamics, in particular what happens at the onset of generalization and why it is so abrupt. Perhaps methods described in Roberts et al. (2021) – where the feature kernel and the neural tangent kernel can be computed analytically throughout the training – will take a particularly simple form in this setting.

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

## A    COMPLEX NETWORK

A simpler network that solves modular addition problem can be phrased using complex weights. This structure would also be more friendly to physicists. The complex solution takes form

$$W_{kn}^{(1)} = \begin{pmatrix} e^{2\pi i \frac{k}{p} n_1 + i\varphi_k^{(1)}} \\ e^{2\pi i \frac{k}{p} n_2 + i\varphi_k^{(2)}} \end{pmatrix}, \qquad n = (n_1, n_2) \tag{20}$$

$$W_{qk}^{(2)} = e^{-2\pi i \frac{k}{p} q - i\varphi_k^{(3)}}, \tag{21}$$

We can take quadratic activation function that simply squares the preactivations. The first preactivation and activation are given by

$$h^{(1)}(n, m) = e^{2\pi i \frac{k}{p} n + i\varphi_k^{(1)}} + e^{2\pi i \frac{k}{p} m + i\varphi_k^{(2)}}, \tag{22}$$

$$z^{(1)}(n, m) = e^{2\pi i \frac{k}{p} 2n + i\varphi_k^{(1)}} + e^{2\pi i \frac{k}{p} 2m + i\varphi_k^{(2)}} + 2 e^{2\pi i \frac{k}{p} (n+m) + i(\varphi_k^{(1)} + \varphi_k^{(2)})}. \tag{23}$$

The final activation is given by

$$h^{(2)}(n, m) = \sum_{k=1}^{N} \left( e^{2\pi i \frac{k}{p} (2n-q) + i(\varphi_k^{(1)} - \varphi_k^{(3)})} + e^{2\pi i \frac{k}{p} (2m-q) + i(\varphi_k^{(2)} - \varphi_k^{(3)})} \right. \tag{24}$$

$$\left. + \quad 2 e^{2\pi i \frac{k}{p} (n+m-q) + i(\varphi_k^{(1)} + \varphi_k^{(2)} - \varphi_k^{(3)})} \right). \tag{25}$$

Similarly setting

$$\varphi_k^{(1)} + \varphi_k^{(2)} - \varphi_k^{(3)} = 0 \tag{26}$$

yields the constructive interference for the output supported on $(n + m - q) = 0 \mod p$.

## B    OTHER ACTIVATIONS

Remarkably, the weights equation 6-equation 7 also solve the modular addition problem for networks equation 1-equation 2 with other activation functions. That is, the function

$$f(x) = \frac{1}{D\sqrt{N}} W^{(2)} \phi \left( W^{(1)} x \right) \tag{27}$$

approximates the $\delta$-function concentrated on the modular addition problem. This also holds for the generalizations discussed in the main text. We do not have an analytic proof of this fact, so we provide the evidence in Fig. 5.

## C    DYNAMICS

In this Section we introduce an empirical measure that quantifies the feature learning for the modular addition task. To define such measure we turn to the exact solution equation 6- equation 7. We will utilize the fact that periodic weights are peaked in Fourier space, while random weights are not.

To define the measure of feature learning, we first transform the weights $W_{nk}^{(1)}$ to a Fourier space with respect to index $n$. Denote the transformed weights $\tilde{W}_{\nu k}^{(1)}$. If the weights are periodic, then Fourier-transformed weights are *localized* in $\nu$, *i.e.* for most values of $\nu$ we have $\tilde{W}_{\nu k}^{(1)} \approx 0$ except for a few values determined by the frequency $\frac{2\pi}{p} k$. At initialization, when the weights are random the Fourier-transformed weights are *delocalized*, *i.e.* will take roughly equal values for any $\nu$.

We introduce a measure of localization known as the inverse participation ratio (IPR). It is routinely used in localization physics Girvin and Yang (2019) as well as network theory Pastor-Satorras and Castellano (2016). We define IPR in terms of the normalized Fourier-transformed weights

$$\text{IPR}_r(k) = \sum_{\nu=1}^{D} |\tilde{w}_{\nu k}^{(1)}|^{2r}, \qquad \text{where} \qquad \tilde{w}_{\nu k}^{(1)} = \frac{\tilde{W}_{\nu k}^{(1)}}{\sqrt{\sum_{\nu=1}^{D} (\tilde{W}_{\nu k}^{(1)})^2}}, \tag{28}$$

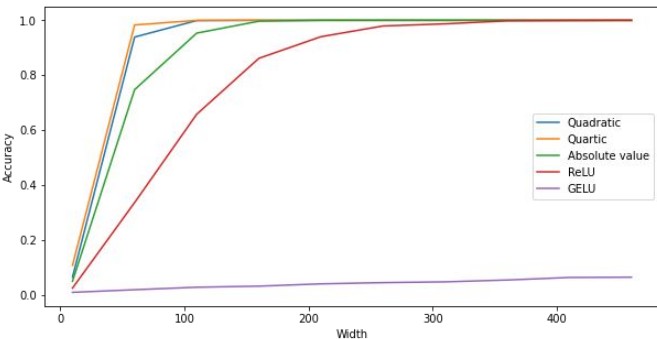

Figure 5: Accuracy for various activation functions. Test accuracy vs. width for different activation functions for $f(n, m) = n + m \bmod p$. The weights are given by equation 6-equation 7. GELU activation eventually reaches $100\%$ accuracy, but at very large width.

and $r$ is a parameter traditionally taken to be 2. It follows from the definition that $\mathrm{IPR}_1(k) = 1$ for any $k$. Unfortunately, $\mathrm{IPR}_r(k)$ is defined per neuron. We would like a single measure for all of the weights in a given layer. Thus, we introduce the average IPR

$$\overline{\mathrm{IPR}}_r = \frac{1}{N} \sum_{k=1}^{N} \mathrm{IPR}_r(k) \,. \tag{29}$$

Larger values of $\overline{\mathrm{IPR}}_r$ indicate that the weights are more periodic, while the smaller values indicate that the weights are more random.

We plot $\overline{\mathrm{IPR}}_2$ as a function of time in Fig. 6. It is clear that there is an upward trend from the very beginning of training. Onset of grokking is correlated with the sharp increase of rate of IPR growth.

## D   SOME OTHER MODULAR FUNCTIONS

The architecture equation 4 can also learn modular multiplication, however in that case we have to convert the products into sums using logarithms over $\mathbb{Z}_p$. We will discuss the modular multiplication elsewhere.

Broadly speaking, a bivariate modular function is a $p \times p$ table where each entry can take values between 0 and $p - 1$. There are $p^{p^2}$ such tables. Grokking is not possible on the overwhelming majority of such functions, because this set includes placing random integers in each entry of the table. Such functions can be represented as modular polynomials of a sufficiently large degree. Some modular functions, namely the ones that involve addition *and* multiplication, *and* are not of the form $\tilde{f}$ are substantially harder to learn. They require more data, more time and do not always yield $100\%$ test accuracy after grokking. One particularly interesting example was found by Power et al. (2022), $f(n, m) = n^3 + nm^2 + m$, which does not generalize even for $\alpha > 0.9$, both for transformer and MLP architectures. Some examples are discussed below. It is not clear how to predict which functions will generalize and which will not given an architecture.

To summarize, a general polynomial of degree 1 is easy to learn and the corresponding MLP is analytically solvable and interpretable, a general polynomial of degree 2 is difficult to learn and is not solvable, while a general polynomial of degree 3 cannot be learnt within the scope of methods we have tried.

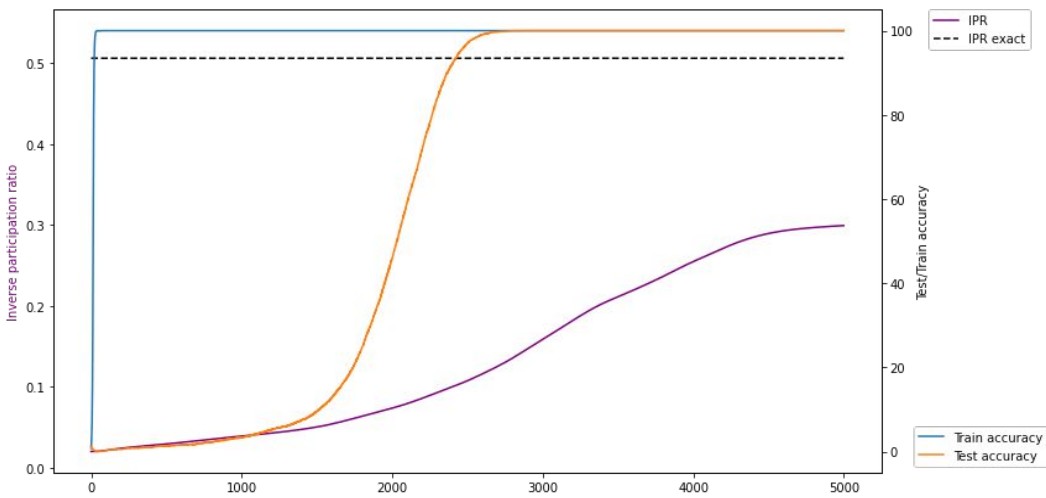

Figure 6: Inverse participation ratio. IPR plotted against the dynamics (under AdamW) of train and test accuracy. Empirically, we see 4 regimes: (i) early training when IPR grows linearly and slowly; (ii) transition from slow liner growth to fast linear growth. This transition period coincides with grokking; (iii) fast linear growth, that starts after 100% test accuracy was reached; (iv) saturation, once weights became periodic. The dashed line indicates $\overline{\mathrm{IPR}}_2$ for the exact solution equation 6-equation 7. The gap between the two indicates that even in the final solution there is quite a bit of noise leading do some mild delocalization in Fourier space. More training and more data helps to reduce the gap.

Next we show a few examples of the modular functions for which the exact solutions discussed in the main text apply.

- $f(n, m) = n^2 + m^2 \bmod p$. Full solution is available and gives $100\%$ accuracy. The first layer weights are given by

$$W_{kn}^{(1)} = \begin{pmatrix} \cos\left(2\pi\frac{k}{p}n_1^2 + \varphi_k^{(1)}\right) \\ \cos\left(2\pi\frac{k}{p}n_2^2 + \varphi_k^{(2)}\right) \end{pmatrix}, \qquad n = (n_1, n_2), \tag{30}$$

  while the second layer weights remain unmodified.

- $f(n, m) = (n + m)^2 \bmod p$. The weights in the first layer are unmodified, while the weights in the second layer are given by

$$W_{qk}^{(2)} = \cos\left(-2\pi\frac{k}{p}q^{\frac{1}{2}} - \varphi_k^{(3)}\right). \tag{31}$$

  Note that $q^{\frac{1}{2}}$ must be understood in the modular sense, that is $r = q^{\frac{1}{2}}$ is a solution to $r^2 = q \bmod p$.

- $f(n, m) = nm$. We do not have an analytic solution. The activations are presented in Fig. 9

- $f(n, m) = n^2 + m^2 + nm \bmod p$. We do not have an analytic solution. This generalization on this function never reaches $100\%$ unless most of the data is utilized, $\alpha > 0.95$. See the learning curve in Fig. 10. Note that although generalization accuracy is very high: $\approx 97\%$, there is a large gap between train and test loss. This is to be contrasted with Fig. 2, where the gap disappears over time.

- $f(n, m) = n^3 + nm^2 + m$. We do not have an analytic solution. The generalization never rises above $1\%$. See the learning curve in Fig. 10.

We show the corresponding activations on Fig. 7 - Fig. 9

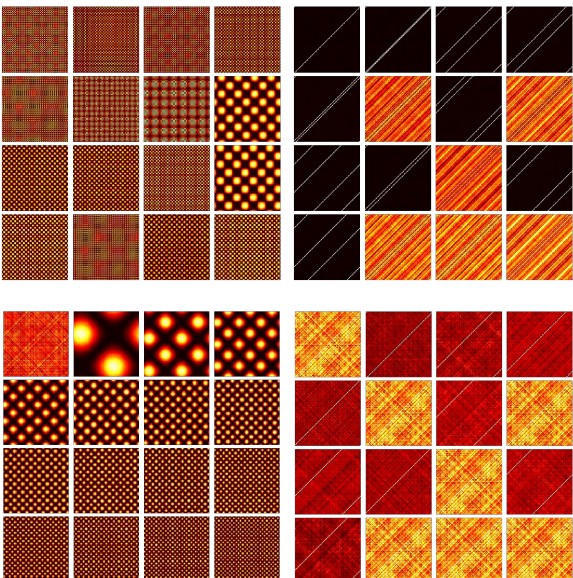

Figure 7: **Top**: Preactivations $h_k^{(1)}$ and $h_q^{(2)}$ found by the AdamW for $f(n, m) = (n + m)^2 \bmod p$. Note that $h_k^{(1)}$ is the same as for $f(n, m) = (n + m) \bmod p$ as expected. **Bottom**: Analytic solution for the same function. Note that since square root is *not* invertible – because it has two branches – the accuracy of analytic solution is $\approx 51\%$. It can be clearly seen in the form of $h_q^{(2)}$: there are $4$ activation lines in the top plots and only $2$ in the bottom. Each pair corresponds to a branch of square root. The noisy preactivations $h_q^{(2)}$ correspond to the values of $q$ that cannot be represented as $(n + m)^2 \bmod p$.

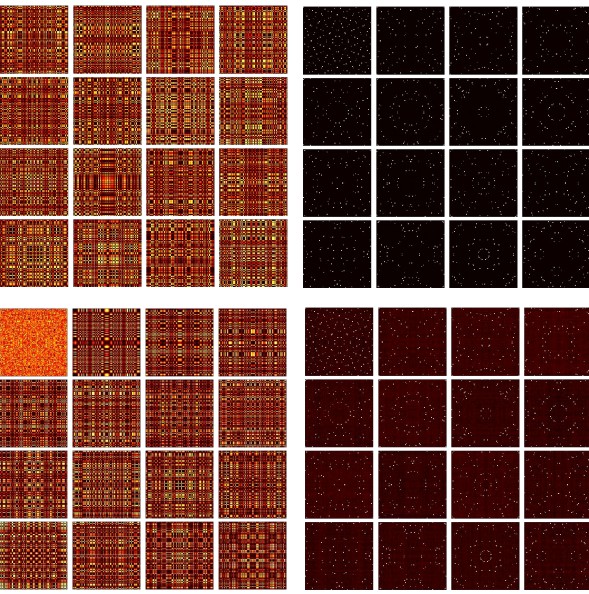

Figure 8: **Top**: Preactivations $h_k^{(1)}$ and $h_q^{(2)}$ found by the AdamW for $f(n, m) = n^2 + m^2 \bmod p$. **Bottom**: Analytic solution for the same function. Both solutions have $100\%$ accuracy.

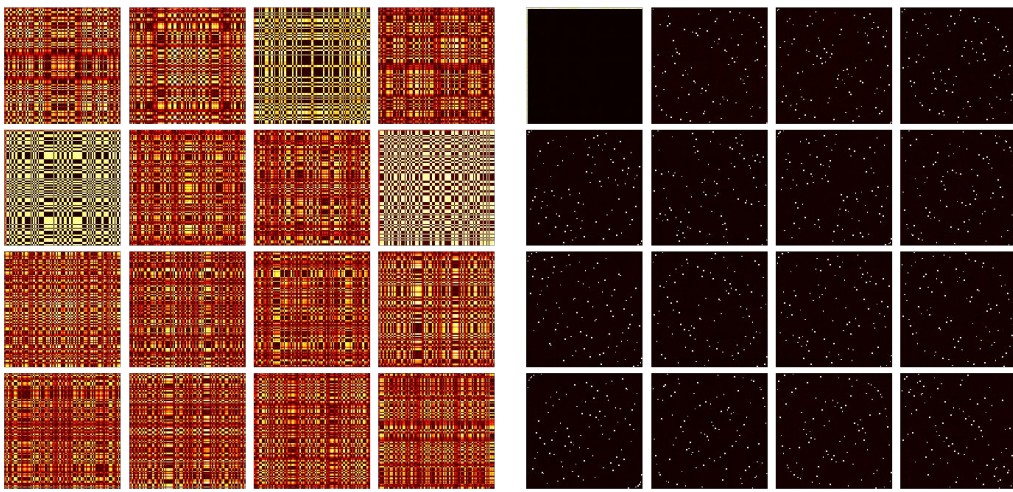

Figure 9: Preactivations $h_k^{(1)}$ and $h_q^{(2)}$ found by the AdamW for $f(n, m) = nm$ mod $p$.

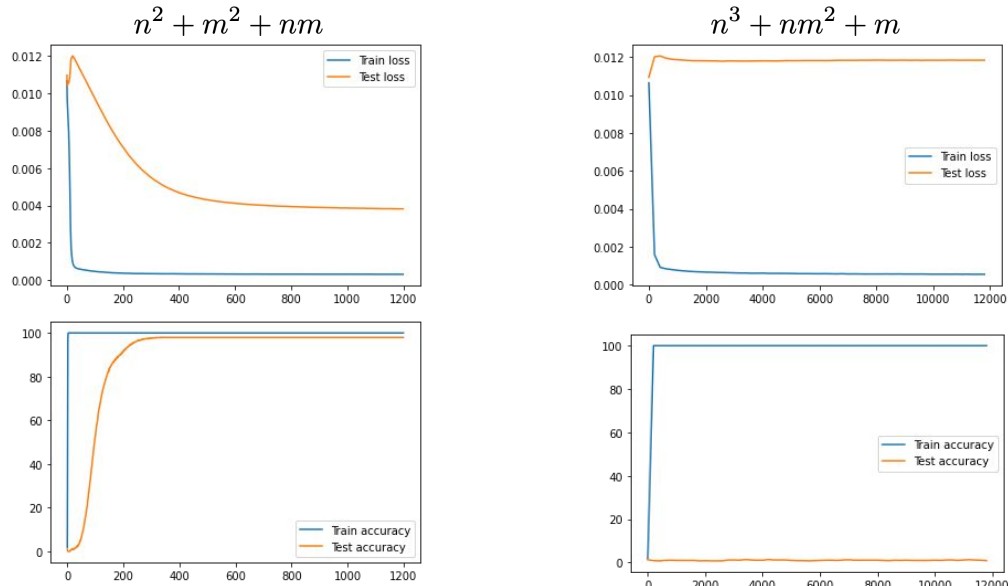

Figure 10: The learning curves for $f(n, m) = n^2 + m^2 + nm$ mod $p$ and $f(n, m) = n^3 + nm^2 + m$ mod $p$ at $\alpha = 0.73$ and $\alpha = 0.9$ correspondingly. Note the gap between train and test loss in the former case. Although test accuracy is almost $100\%$, it is clear that the network did not grok all the right features.

# E  ATTENTION AND TRANSFORMERS

In this Section we empirically investigate grokking in a single-layer Transformer as well as the pure attention layers.

Grokking was initially discovered in two-layer transformerPower et al. (2022), while the periodic structure in MLP activations was found in the case of a single-layer transformer Nanda and Lieberum (2022). Here we slightly simplify the setup by using the MSE loss (showing in passing that slingshots Thilak et al. (2022) are not necessary for the generalization to occur).

We reproduce the periodicity of the MLP activations, see Fig. 11 as expected. We also note that embedding and un-embedding matrices generically do not develop structure in Fourier space. This indicated that periodic structure is universal, while it's realization in terms of weights is not. This further underlines the utility of the analytic solution described in the present work.

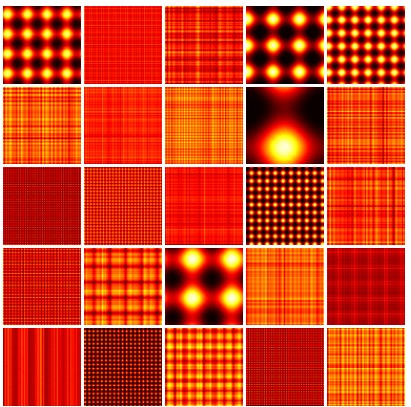

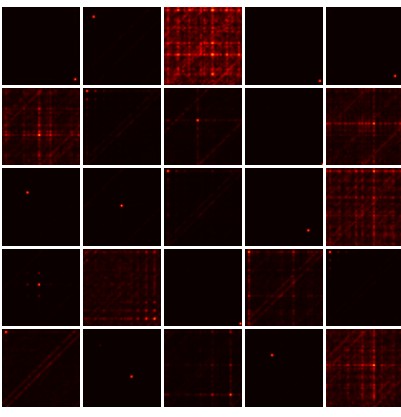

(a) Activations in the MLP layer

(b) Activations in the MLP layer in Fourier space

Figure 11: Activations (and their Fourier transforms) in the MLP layer of a single-layer transformer architecture. Note that the activations are periodic exactly the same way as in the MLP case, while their realization in terms of weights is different.

### E.2 ATTENTION-ONLY "TRANSFORMER"

Based on the main body of the paper, it is tempting to conclude that the computations relevant for modular addition are always done in the MLP layers. We are going to challenge this conclusion in the present Section by removing the MLP layers from the transformer architecture. Indeed, in the main text we showed that MLP alone can solve the modular arithmetic problems. Consequently, adding more non-linearity via attention is not necessary: if anything, attention only gets on the way making the optimization more difficult. In all cases when MLP is present periodic activations appear there. Now we will ask the opposite question: can attention alone, being a non-linear operation, ensure generalization and grokking on arithmetic tasks?

Experiments reveal that when generalisation occurs, the dynamics shows the same characteristic features as MLP discussed in the main text and Transformers discussed in the literature as well as the present section. Fig.12.

We make, however, a surprising observation: generalization on modular addition is only possible if enough attention heads are present. There is a headcount-driven grokking transition at *fixed* number of parameters. This transition is present in a single-layer linear and softmax Attention, as well as in two-layer softmax Attention.

We present the results in Fig. 13, where the final train/test accuracy[4] are plotted as functions of number of heads. The experiments are done for $p = 97$ and $n_{embd} = 128$. It is clear from Fig. 13

---

[4]With early stopping used if needed

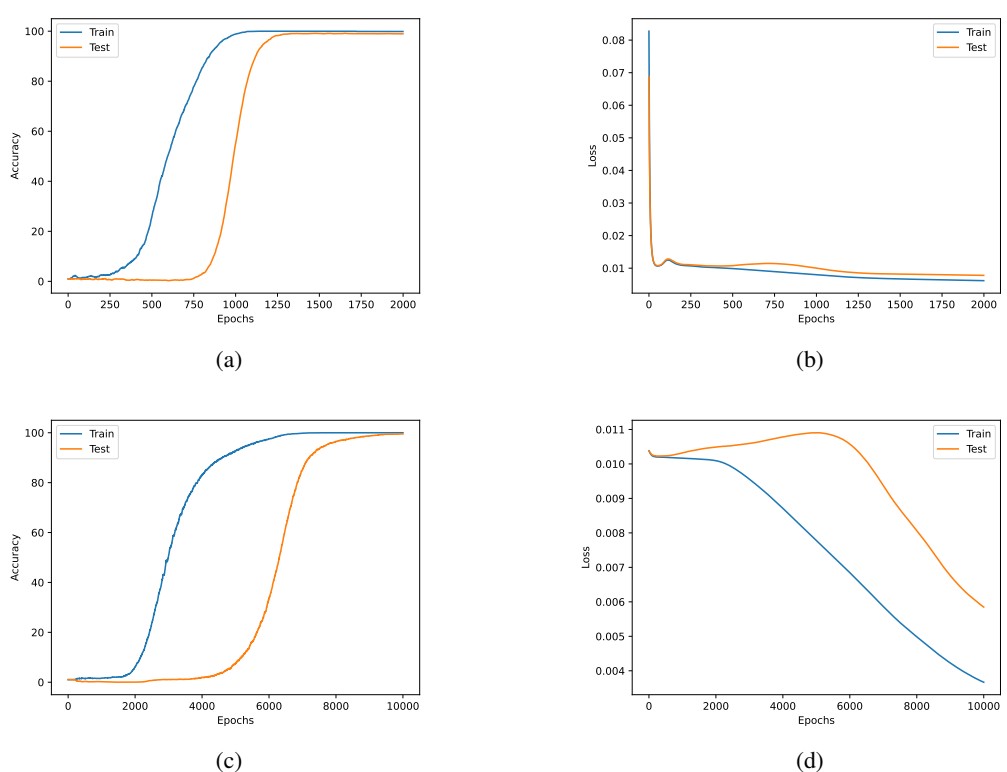

Figure 12: (a),(b): Accuracy and Loss for linear attention layers on modular addition. (c),(d): Accuracy and Loss for Softmax attention. In both cases dynamics shows the same signatures as in the MLP case.

that when the number of heads is small, Attention-only network fails to approximate the training set well.

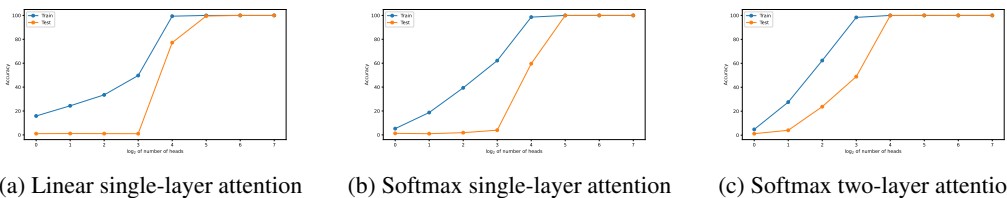

(a) Linear single-layer attention     (b) Softmax single-layer attention     (c) Softmax two-layer attention

Figure 13: In all cases we observe a generalization jump driven by the number of heads. In the experiments the number of parameters is held fixed and only the number of heads (and, correspondingly, head size) are changed.

