# OpenReview forum: "A simple and interpretable model of grokking modular arithmetic tasks"
_ICLR.cc/2024/Conference — Submitted to ICLR 2024_

### Official Review · Reviewer_QK7D · 2023-10-30

**Soundness:** 2 fair
**Presentation:** 2 fair
**Contribution:** 2 fair
**Rating:** 5
**Confidence:** 3

**Summary:**

This manuscript introduces a simple setup to reproduce the  grokking phenomenon on modular arithmetic problems. Different from existing works, the major contribution is the authors provide an  analytic solutions for  two-layer quadratic networks of solving modular arithmetic problems. Additionally, the authors show that in experiments, typical algorithms like SGD and Adam indeed find solutions that resemble the analytic ones.

**Strengths:**

- The proposed setup  is simple and interpretable.
- The analytic solutions could be valuable in analyzing the grokking phenomenon for the tasks of modular arithmetic.

**Weaknesses:**

- The constructed analytic solutions for the tasks of modular arithmetic has potential but specifically, this manuscript does not produce too much new insights for understanding grokking. For instance, one can easily construct analytic solutions for learning k-sparse parity with two-layer ReLU networks, where we can reproduce the grokking phenomenon.

- The authors have empirically shows a peak around 0 for $\phi_k^{(1)} + \phi_k^{(2)} - \phi_k^{(3)}$ in the found solution, satisfying equality they propose. However, the presentation falls short of providing adequate evidence that the found weights have the periodic structure of the analytic solution. It is imperative that the authors supplement their work with further empirical evidence or a comprehensive theoretical analysis to elucidate how the weights progressively evolve toward the analytic solution during the training process.
- Further investigation into the minimal data amount of grokking occurrences is warranted. Does the order of the minimal amount is $O(p^2)$. If not, it necessitates a more suitable definition of the fraction as presented in equation (3).
- Some mathematical oversight.
   - The definition in (6) might lead to the misconception that the weights $W_{kn}^{1}$ form an $N \times p^2 $ matrix
   - The  factor $\frac{1}{N}$ is missing at the beginning of (11).

**Questions:**

None

---

> ### Author Response · Authors · 2023-11-21
> **Response**
>
> I thank the reviewer for positive comments and questions!
>
> **Strengths:**
> _The proposed setup is simple and interpretable.
> The analytic solutions could be valuable in analyzing the grokking phenomenon for the tasks of modular arithmetic._
>
> **Weaknesses:**
>
> _The constructed analytic solutions for the tasks of modular arithmetic has potential but specifically, this manuscript does not produce too much new insights for understanding grokking. For instance, one can easily construct analytic solutions for learning k-sparse parity with two-layer ReLU networks, where we can reproduce the grokking phenomenon._
>
> I am not familiar with the result. I suspect that either solution is not found by optimization (of which there are examples) or only makes sense for ReLU networks.
>
> To further elaborate, the solution I presented works for arbitrary number of variables and over $\mathbb{Z}_p$ with any $p$. $k$-sparse parity is a particular case of this setting with $p=2$ and linear modular function with $k$ non-zero coefficients. I would be very interested to compare the two solutions if the reviewer can provide a reference where such solution for ReLU network is constructed.
>
> _The authors have empirically shows a peak around 0 for
>  in the found solution, satisfying equality they propose. However, the presentation falls short of providing adequate evidence that the found weights have the periodic structure of the analytic solution. It is imperative that the authors supplement their work with further empirical evidence or a comprehensive theoretical analysis to elucidate how the weights progressively evolve toward the analytic solution during the training process._
>
>  There must have been a poor explanation on my part: The very notion of a phase is not meaningful _unless_ the weights are periodic. Consequently, the very presence of the phases implies that the rows of the weight matrix are periodic. The phases are extracted by fitting the rows of the weight matrices with $\cos(\omega + \phi)$ and then extracting $\phi$.
>
>  I further demonstrate the periodicity in the Supplementary Material, Section C on dynamics (see general comments above and Fig.6 of Supplementary material), where inverse participation ratio (IPR) of the Fourier-transformed weights is defined and measured as a function of time. Growth of IPR indicates that weights become increasingly more periodic (_i.e._ more localized in Fourier space) over time.
>
> _Further investigation into the minimal data amount of grokking occurrences is warranted. Does the order of the minimal amount is $O(p^2)$
> . If not, it necessitates a more suitable definition of the fraction as presented in equation (3)._
>
> The equation (3) defines an $\alpha$ which is simply a "dimensionless" (meaning independent of $p$) measure of how much data was used for training. It is a  hyperparameter that is independent of $p$ by definition. The _minimal_ amount of training data $\alpha_c$ does weakly depend on $p$ and slowly decreases as $p$ gets larger. This effect has been observed in [1] where linear functions of many variables over $Z_p$ are learnt and larger $p$ lead to "simpler" learning task.
>
> **References:**
>
> [1] Emily Wenger, Mingjie Chen, François Charton, Kristin Lauter, ``SALSA: Attacking Lattice Cryptography with Transformers''
>
>
> **Some mathematical oversight.**
>
> _The definition in (6) might lead to the misconception that the weights form an matrix $Nxp^2$._
>
> Fixed!
>
> _The factor $\frac{1}{N}$ is missing at the beginning of (11)._
>
> I thank the reviewer for their diligence! This choice was deliberate to lighten the equations. I state: ``(we drop the normalization factors)'' above Eq.(8) and restore the $\frac{1}{N}$ factor in Eq.(16).
>
> The $\frac{1}{N}$ factors are not critical, unless we wish to have a finite large-$N$ limit, which is the normalization I opted for in the end.

---

### Official Review · Reviewer_5GzG · 2023-11-02

**Soundness:** 4 excellent
**Presentation:** 4 excellent
**Contribution:** 4 excellent
**Rating:** 8
**Confidence:** 4

**Summary:**

This paper studies the gokking phenomenon by fitting two-layer MLP on modular arithmetic tasks. The paper obtains explicit periodic features in the solutions, and shows that gokking occurs when the correct features are learned.

**Strengths:**

This paper is a timely and important contribution to the growing literature on gokking. It offers a class of problems with explicit solutions, so that gokking can be studied in great depth.

**Weaknesses:**

While simplicity and explicit solution are a strength, it also limits the scope of the paper in terms of covering the gokking phenomenon in general. Moreover, it is desirable to study the dynamics of the optimizers in reaching the exact solutions, but the paper did not make such an attempt.

**Questions:**

The transition from memorization to generalization appears to be a continuous process of Occam's razor, i.e., gradually reducing the complexity of the model while maintaining the training error. Converging to periodic features is also of this nature. Is this the correct understanding of gokking?

---

> ### Author Response · Authors · 2023-11-21
> **Response**
>
> I thank the reviewer for the positive comments and questions!
>
> **Strengths:**
>
> _This paper is a timely and important contribution to the growing literature on grokking. It offers a class of problems with explicit solutions, so that grokking can be studied in great depth._
>
> **Weaknesses:**
>
> _While simplicity and explicit solution are a strength, it also limits the scope of the paper in terms of covering the grokking phenomenon in general. Moreover, it is desirable to study the dynamics of the optimizers in reaching the exact solutions, but the paper did not make such an attempt._
>
> I have addressed both of these questions in the general comments part of my response.
>
> I added experiments and some discussion with other architectures: Transformers and pure attention layers (please see general comments for detailed description of those).
>
> Discussion of the dynamics appears in the Supplementary material. In the case of modular addition, the network starts moving towards a periodic solution right away (as I show by measuring inverse participation ratio of Fourier-transformed weights over time) and this motion accelerates after a certain point, at this point generalization picks up.
>
> More details about the dynamics are the subject of work in progress.
>
> **Questions:**
>
> _The transition from memorization to generalization appears to be a continuous process of Occam's razor, i.e., gradually reducing the complexity of the model while maintaining the training error. Converging to periodic features is also of this nature. Is this the correct understanding of grokking?_
>
> One important motivation of the work was to flesh out the final solution in as much detail as possible to address this very question. I am not aware of any quantitative (meaning computable) measure of complexity that monotonically decreases over the course of training for all learnable modular functions and all hyperparameter and initialization choices. In particular, sparsity in Fourier space (which is only relevant for addition) or weight norms (which increase according to Fig.1 of my work) are _not_ relevant measures. Consequently, I am skeptical of explanations based on any notion of complexity.
>
> The best I can presently offer is: for sufficient amount of data ($\alpha > \alpha_c$) most low training loss minima are generalizing, while for insufficient amount of data $(\alpha < \alpha_c)$ most low training loss minima are memorizing. The critical amount of data $\alpha_c$ depends strongly on the modular operation we are trying to learn and weakly on the optimizer. _How_ this transition happens is not clear and is an exciting research direction.

---

> > ### Comment · Reviewer_5GzG · 2023-11-22
> > **Thanks for answering my question**
> >
> > I think this paper should be accepted.

---

### Official Review · Reviewer_PaA4 · 2023-11-02

**Soundness:** 2 fair
**Presentation:** 3 good
**Contribution:** 2 fair
**Rating:** 6
**Confidence:** 3

**Summary:**

The authors present a two-layer MLP for solving modular arithmetic tasks. The goal is to study a sudden jump in generalization during training, known as grokking. An analytic solution of the model weight is derived, guaranteeing 100% test accuracy. A general result for  arithmetic addition is also given. The experiments show that the proposed representation is also found by training using gradient descent and AdamW.

**Strengths:**

- The analysis of grokking help to understand and dynamics of model training and how to achieve good generalization
- The theoretical results are applicable for general modular functions. Follow-up work could leverage on these results.

**Weaknesses:**

- Simple architecture and tasks (two layer MLP, modular arithmetic) could limit the applications and extensions of this work
- The given analytical solution does not help much in understanding how grokking happens as the latter occurs earlier than achieving 100% test accuracy.

**Questions:**

- How does the analytical solution help understanding grokking ?
- Neural networks are known to converge to local minima. I wonder if there are potentially other analytical solutions and why it seems that model training leads to the same solution.

---

> ### Author Response · Authors · 2023-11-21
> **Response**
>
> I thank the reviewer for the positive comments and questions!
>
> **Strengths:**
> _The analysis of grokking help to understand and dynamics of model training and how to achieve good generalization
> The theoretical results are applicable for general modular functions. Follow-up work could leverage on these results._
>
>
> **Weaknesses:**
> _Simple architecture and tasks (two layer MLP, modular arithmetic) could limit the applications and extensions of this work
> The given analytical solution does not help much in understanding how grokking happens as the latter occurs earlier than achieving 100$\%$ test accuracy._
>
> I have addressed some of these concerns in the general comments above. I have added experiments with transformers and with pure attention layers, and described the results above.
>
> Part of this question is addressed in the Supplementary Material on dynamics (also, see general comments above). In the case of modular addition the network starts moving towards a periodic solution right away (as I show by measuring inverse participation ratio of Fourier-transformed weights over time) and this motion accelerates after a certain point, at this point generalization picks up. The progress measure -- inverse participation ratio -- is motivated by the analytic solution: it measures periodicity of the weights.
>
> More details about the dynamics are the subject of the work in progress.
>
> **Questions:**
> _How does the analytical solution help understanding grokking ?
> Neural networks are known to converge to local minima. I wonder if there are potentially other analytical solutions and why it seems that model training leads to the same solution._
>
> In the present work I could only answer where the dynamics converges to. The questions why and how are extremely complicated even in the case of the simplest possible model. Detailed understanding of the dynamics is still work in progress and will appear in a separate paper.
>
> There are memorizing solutions, however I am not aware of any other generalizing solutions.

---

### Official Review · Reviewer_UC6H · 2023-11-02

**Soundness:** 3 good
**Presentation:** 3 good
**Contribution:** 2 fair
**Rating:** 5
**Confidence:** 4

**Summary:**

Using a two-layer MLP, this paper analyzes the phenomenon of grokking on a few modular arithmetic problems. Due to the simple DNN architecture, the weights and features are calculated analytically to solve modular addition problems to provide mechanical details about what was learned by the model.

**Strengths:**

1. Grokking is an interesting and exciting phenomenon that is worth careful study.
2. The paper is technically sound.
3. The presentation and organization is clear.

**Weaknesses:**

1. To provide an analytical solution and interpretability of the model, this paper focuses on a very simple model (definitely not used in practice) and arithmetic function to be learned, which limits its impact on practical models currently in use, such as CNN, Transformer.

2. If the model really learns the arithmetic function, it will be interesting to see whether the model generates accurate results for OOD data, e.g., training with the data from [0, 10], testing with the data from [1000, 1100].

**Questions:**

1. "Instead, large width leads to redundant representations: Each frequency appears several times with different random phases ultimately leading to a better wave interference" Since they are identical frequencies, will combining them provide a more concise representation?

2. Besides the amount of data, how is grokking affected by the training data distribution?

---

> ### Author Response · Authors · 2023-11-21
> **Response**
>
> I thank the reviewer for the positive comments and questions!
>
> **Strengths:**
> _Grokking is an interesting and exciting phenomenon that is worth careful study.
> The paper is technically sound.
> The presentation and organization is clear._
>
> **Weaknesses:**
> _To provide an analytical solution and interpretability of the model, this paper focuses on a very simple model (definitely not used in practice) and arithmetic function to be learned, which limits its impact on practical models currently in use, such as CNN, Transformer._
>
> Some of the concerns are addressed in the general comments above.
>
> I have addressed this comment by making further experiments with transformers and pure-attention networks. Initially, I chose not to include these architectures because grokking was first empirically observed in two-layer transformers, while the blogpost [https://www.neelnanda.io/blog/interlude-a-mechanistic-interpretability-analysis-of-grokking] empirically considered one-layer transformer.
>
> My objective was to distill the _simplest possible model_ of the phenomenon rather than focus on models that are more realistic, but cannot be understood in as much detail. Since the identical effects (jump in generalization as a function of time and amount of data) are present in all cases -- simple and complex --, it is reasonable to make as much progress as possible in the simplest case. (Such research philosophy has been very successful in theoretical physics so far).
>
>
> _If the model really learns the arithmetic function, it will be interesting to see whether the model generates accurate results for OOD data, e.g., training with the data from [0, 10], testing with the data from [1000, 1100]._
>
> One advantage of the analytic solution is that the reviewer does not need to take my word on it! The learnt feature maps are described by equations and are made from trigonometric functions. In the main text I prove _analytically_ that these features perform modular operations including addition and show empirically that similar features are learnt during optimization.
>
> The model learns _modular_ arithmetic. That is, arithmetic over a finite field $\mathbb{Z}_p$ rather than over $\mathbb Z$. The modulus $p$ appears explicitly in the analytic solution and in the dimension of the input and output of the architecture. In the transformer case, $p$ will appear as vocab_size, unless a representation of integers with base $B<p$ is used (in which case ``tokenization scheme'' is determined by the pair $(B,p)$). If numbers larger than $p$ are presented to the model, it will round them down modulo $p$.
>
> Meta learning many different operations in one network is beyond the scope of this paper.
>
> **Questions:**
> _"Instead, large width leads to redundant representations: Each frequency appears several times with different random phases ultimately leading to a better wave interference" Since they are identical frequencies, will combining them provide a more concise representation?_
>
> Unfortunately, combining the neurons with same frequencies is not possible because in addition to frequency the trigonometric functions contain a random phase, which is different for different neurons. To be concrete, neurons with $\cos(\omega + \phi_1)$ and $\cos(\omega + \phi_2)$ cannot be easily replaced by a single neuron.
> What is possible, however, is to simply keep erasing the ``redundant'' neurons. This will keep performance as measured by accuracy constant (up to some point), but will decrease the train and test loss.
>
> _Besides the amount of data, how is grokking affected by the training data distribution?_
>
> This questioned was mentioned as open in the paper: ``The value of $\alpha_c$ further depends on how training set is sampled from the entire dataset; the appropriate choice of the sampling method may thus improve the data efficiency.''
>
> Unfortunately, I had very little progress in this direction. Empirically it appears that sampling the finite dataset with the uniform probability gives the best performance. We did some experiments along the lines of work [1], but no benefit was seen.
>
> **References:**
>
> [1] Mansheej Paul, Surya Ganguli, Gintare Karolina Dziugaite, ``Deep Learning on a Data Diet: Finding Important Examples Early in Training''

---

> > ### Comment · Reviewer_UC6H · 2023-11-22
> > **Thanks for the authors' response.**
> >
> > Based on the authors' response, my rating remains the same.

---

### Official Review · Reviewer_9nDi · 2023-11-08

**Soundness:** 4 excellent
**Presentation:** 3 good
**Contribution:** 4 excellent
**Rating:** 6
**Confidence:** 3

**Summary:**

This paper studies the problem of learning modular arithmetic with a two-layer network. It proposes a certain Ansatz for the final weights based on Fourier analysis and experimentally shows that the weights match this Ansatz.

**Strengths:**

The paper's presentation is clear and to the point, and the construction of the weights is succinctly explained. The experimental evidence is convincing. Mechanistic interpretability is also a highly interesting direction overall.

**Weaknesses:**

1) Literature review is missing some recent work:

* There is a growing body of work on learning single-index and multi-index functions (see e.g., "Online stochastic gradient descent on non-convex losses from high-dimensional inference" by Ben Arous et al., and "SGD learning on neural networks: leap complexity and saddle-to-saddle dynamics" by Abbe et al.) which shows similar grokking effects. It could be interesting to understand how these relate to the arithmetic grokking effect.

* More crucially, there was a paper called "Progress measures for grokking via mechanistic interpretability" which appeared online in Jan., 2023 and was published in ICLR 2023. This paper also seems to derive the Fourier-based solution to the grokking task. This seems unfortunate, because it seems that at the time that this paper was written either the authors are unaware of this other paper, or that this paper was written concurrently and has been to a good extent subsumed by that other paper. Could the authors comment on this? This is the main weakness in my mind.

2) The analysis only gives an Ansatz for the final solution of the weights, but does not explain why more/less data leads to finding it, and why there is a sharp jump in the algorithm's loss from not finding the Ansatz to finding the Ansatz. In other words, the paper only predicts the final weights but does not give an interpretation of what is driving the dynamics of the grokking process.

**Questions:**

1. What is meant by "Functions of the form f(n, m) + g(n, m) mod p are more difficult to grok: they require more epochs and larger α"? Why do you need both f(n,m) and g(n,m) here?
Typos:
"a lightening" -> "an enlightening"
"we do not observer"

---

> ### Author Response · Authors · 2023-11-21
> **Response**
>
> I thank the reviewer for the positive comments on my work and questions!
>
> **Strengths:**
> _The paper's presentation is clear and to the point, and the construction of the weights is succinctly explained. The experimental evidence is convincing. Mechanistic interpretability is also a highly interesting direction overall._
>
> **Weaknesses:**
> _Literature review is missing some recent work:
> There is a growing body of work on learning single-index and multi-index functions (see e.g., "Online stochastic gradient descent on non-convex losses from high-dimensional inference" by Ben Arous et al., and "SGD learning on neural networks: leap complexity and saddle-to-saddle dynamics" by Abbe et al.) which shows similar grokking effects. It could be interesting to understand how these relate to the arithmetic grokking effect._
>
> I was not aware of that line of work. Presently, it is not clear to me what the relation is. The point I make in my paper is that grokking is tightly connected to feature learning. The exact shape of the learning curve is not sufficient. I now cite this work in the new version.
>
>
> _More crucially, there was a paper called "Progress measures for grokking via mechanistic interpretability" which appeared online in Jan., 2023 and was published in ICLR 2023. This paper also seems to derive the Fourier-based solution to the grokking task. This seems unfortunate, because it seems that at the time that this paper was written either the authors are unaware of this other paper, or that this paper was written concurrently and has been to a good extent subsumed by that other paper. Could the authors comment on this? This is the main weakness in my mind._
>
> My work appeared online **before** "Progress measures for grokking via mechanistic interpretability" paper and, consequently, I opted not to cite it in the present version. I have now added the citation with a comment on the timeline.
>
> I would like to further emphasize that the (very interesting) paper "Progress measures for grokking via mechanistic interpretability" (2301.05217v1), which appeared _after_ mine, **does not contain the analytic solution I have described in my work.** The tables with trigonometric identities are obtained empirically and serve as interpretation of the activations measured in experiment. The authors of 2301.05217 are aware of my work, but choose to not cite it.
>
> _The analysis only gives an Ansatz for the final solution of the weights, but does not explain why more/less data leads to finding it, and why there is a sharp jump in the algorithm's loss from not finding the Ansatz to finding the Ansatz. In other words, the paper only predicts the final weights but does not give an interpretation of what is driving the dynamics of the grokking process._
>
> The current version of the paper concludes: ``Given the simplicity of our setup, the basic explanation for grokking must be quite banal: At some point in training (and assuming there is enough data), the only way to decrease the training loss is to start learning the right features.''. This is my current opinion on the dynamics: all low training loss minima generalize if there is enough data.
>
> I do, however, completely agree with the reviewer: there is no clear picture for the fraction of the data and for what exactly changes when this fraction crosses its critical value. It is still work in progress.
>
> **Questions:**
> _What is meant by "Functions of the form f(n, m) + g(n, m) mod p are more difficult to grok: they require more epochs and larger $\alpha$"? Why do you need both f(n,m) and g(n,m) here?_
>
> Indeed, more difficult means more epochs and more data. Two examples are shown in Fig. 10 of the Supplementary Material.
> The form $f(n, m) + g(n, m) \quad \textrm{mod} \quad p$ is a slight abuse of notation meant to indicate ``functions of $n$ and $m$ that cannot be written as $F(f_1(n) + f_2(m))$''. To be concrete, $n^2 + m^2 + 2nm$ is not in this class, while $n^2 + m^2 + nm$ is.
>
> _Typos: "a lightening" -> "an enlightening"_
>
> This is indeed a typo, however I meant a more humble "a lightning"! Fixed now!
>
> _"we do not observer"_
>
> Fixed!

---

> > ### Comment · Reviewer_9nDi · 2023-11-22
> >
> > > I was not aware of that line of work. Presently, it is not clear to me what the relation is.
> >
> >
> > I actually think that the works are quite similar in a qualitative sense! Networks learn in the multi-index function setting with a grokking phenomenon that accompanies feature learning. In that setting, it has been proved that the neural network training plateaus and then suddenly experiences a sharp drop. This sharp drop in the loss occurs after a predictable number of samples, when feature learning occurs. In that case, feature learning is when weights align with the relevant subspace of the input.
> >
> >
> > > My work appeared online before "Progress measures for grokking via mechanistic interpretability" paper and, consequently, I opted not to cite it in the present version. I have now added the citation with a comment on the timeline.
> >
> > Since this is an anonymous review process, I of course cannot confirm that your work appeared earlier than the arXiv version -- perhaps the Area Chair can do this? I'm unsure how this can be verified.
> >
> > > I would like to further emphasize that the (very interesting) paper "Progress measures for grokking via mechanistic interpretability" (2301.05217v1), which appeared after mine, does not contain the analytic solution I have described in my work.
> >
> > I also apologize for my oversight in my original review:  indeed, the analytical expression derived in this paper does not appear in  "Progress measures for grokking via mechanistic interpretability". Accordingly, I will raise my score to 8 (accept), since this paper has a novel contribution on an interesting topic.

---

> > > ### Comment · Reviewer_9nDi · 2023-11-23
> > >
> > > Apologies for the back-and-forth, but I felt quite confused again and wanted to make sure that I understand correctly the situation.
> > >
> > > I looked at the Nanda-Lieberum blogpost as it appeared in August 2022 via the wayback machine: https://web.archive.org/web/20220817015241/https://www.alignmentforum.org/posts/N6WM6hs7RQMKDhYjB/a-mechanistic-interpretability-analysis-of-grokking
> > >
> > > In this blogpost the observation that the networks implement the Discrete Fourier transform is stated. The equations for the Discrete Fourier transformer are given. There is a section on Modular Addition where the network is reverse-engineered.
> > >
> > > After reading this blog post more carefully, it seems that the principal contribution of the present manuscript is in Section 3, which writes an explicit set of the weights under which the network implements the Fourier transform and shows that the quadratic activation works. The main difference with the Nanda-Lieberum blogpost is that the weights are written explicitly and the quadratic activation is shown to work.
> > >
> > > After this closer analysis, I am moving my score to 6 (between my original 5 and my current 8), since I find the problem studied in this paper valuable, the solution valuable, but think that the contributions beyond the Nanda-Lieberum blogpost are somewhat limited in scope.

---

> > > > ### Author Response · Authors · 2023-11-23
> > > > **Response**
> > > >
> > > > The reviewer's understanding is correct, and I genuinely appreciate the diligence. The blogpost is properly cited in the original version of my manuscript and the relation of the blogpost to my work is also explained in a dedicated footnote (which I added for ICLR).
> > > >
> > > > I also would encourage the Reviewer to be more careful when referring to "network is reverse-engineered". The mechanism of _how_ the "reverse engineered algorithm" works --- via destructive and constrictive interference due to choice of phases --- does **not** follow from the material presented in the blogpost. In fact, the phases of the periodic activations are not studied. (The word "phase" appears many times, but in a different the context of phase change/transition). The conclusion that Grokking is due to limited data + regularization is not right as I showed by getting the effect experimentally without weight decay and with full-batch GD. Weight decay helps, but explanation of grokking cannot rely on weight decay because it is not mandatory.
> > > >
> > > > I would like to add, although this may appear as a trivial extension to the reviewer, that in my work I have described solutions not only for addition, but a variety of other operations, all belonging to the same **easy** complexity class. Examples of these tasks include functions like $x^2 + y^4$ or $(x+y)^2$. Activations/weights for such functions are **not** periodic (a few examples can be found in the Supplementary material Section on ``Other modular functions'') and, consequently, the Fourier transform is not relevant. Yet, the weights can be determined analytically. Consequently, the computation performed inside the network is not inherently related to DFT. Why optimization chooses the cosines combined with nonlinear operations (such as cos(x^2)) is presently an open problem.

---

### Author Response · Authors · 2023-11-20
**General comments for all Reviewers**

I thank the reviewers for carefully reading the manuscript as well as for their suggestions and questions. The reviewers jointly agree that my work is (i) correct, (ii) provides a useful, simple and solvable model of an MLP network after grokking has occurred (iii) timely, and (iv) on the topic of great interest to the community. I hope that the improved version of the manuscript along with additional experiments and explanations meets their standards. In this part, I will focus on the questions raised by multiple reviewers, while in the next part I will address specific questions asked by the reviewers in detail.

First, however, I would encourage the reviewers to take a slightly different look at the paper. It addresses two main questions. (i) What is necessary and not necessary for grokking to occur? It turns out that weight decay, CSE loss, sharp weight norm spikes, gradient noise, flatter minima, or transformer architecture are not necessary. Any form of non-linearity and capacity for feature learning are necessary. My work is the first to narrow it down to a very simple model. (ii) How universal is the solution found by optimization? The universality is demonstrated by comparing empirically obtained networks to the analytic solution.
At the conceptual level, it should be compared to the Ising model of magnetism in statistical physics: On the one hand, the Ising model does not describe real magnets (it is just 0s and 1s on a square 2D grid). On the other hand, it allows complete analytic understanding of the phase transition between ordered ferromagnetic (cf generalizing solution) and unordered paramagnetic (cf memorizing solution) phases.

The two main themes that arose in the reports are:

**The simple solution applies to the simple architecture. What can be said about other architectures such as transformers?**

To address these concerns I performed experiments with (i) single layer transformer and (ii) single layer attention. These results have been added to the Supplementary material.

In the blogpost [1] the transformer experiments were done and periodicity in (the MLP part of the network) activations was observed. I reproduced the result with MSE loss and plotted the activations in the same way as in the main text. See Fig. 11 and Subsection E.1. These experiments emphasize that although the realization of the network function in terms of weights is more complicated and cannot be described analytically, the network performs essentially the same computation at the level of activations as in the MLP case. This further underlines the utility of the analytic solution I constructed.

I also wanted to show that MLP itself is not mandatory for grokking to occur. Consequently, in the Subsection E.2 I removed the MLP layers from the transformer architecture leaving pure attention networks. Pure attention architecture was not studied in the context of modular arithmetic or other algorithmic datasets, to the best of my knowledge. In this case, I found a surprising **grokking transition driven by the number of heads** (keeping the total number of parameters and the amount of data fixed). This transition is present in both linear and Softmax versions of attention, and, to the best of my knowledge has not been discussed previously.

**The dynamics is a critical part of the story. Why is it not addressed in the manuscript?**

The original version of the manuscript discussed the dynamics, however I had to move it into Supplementary material due to ICLR space constraints.
In the current version of the manuscript available to the reviewers there is an unfortunate typo in the .tex file that placed the name of the section, "Dynamics", inline. See the Section "Other activations" for the complete text (above Eq.(28)). I have now fixed the typo and Dynamics is discussed in the Section C.

In that section I leveraged the analytic solution for modular addition to introduce a measure of feature learning called inverse participation ratio or IPR (Eq.28). It characterizes how periodic the weights are (by quantifying the degree of localization in Fourier space) I, then, measured it over time and showed that it grows first slowly and then rapidly (Figure 6). IPR is an example of what is sometimes called "hidden progress measure". It measures how periodic the weights are on average, and therefore tells us how close the solution is to the analytic one. Some neurons always end up stuck and do not participate in the computation, leaving the IPR large, but not as large as for the analytic solution.

This, by no means, solves the question of the how exactly the sudden jump in generalization happens dynamically, however it does show that periodicity starts to develop right away (although it is very slow at first). More detailed understanding of the dynamics is currently a work in progress and will be delegated to a separate publication.

---

### Meta-Review · Area_Chair_HLci · 2023-12-05

**Metareview:**

This paper is well written and clear, showcases grokking phenomena, and provides compelling closed form solutions. While the original submission was focused on MLP architecture, the author(s) have added additional experiments during the rebuttal period with extensions to transformer architecture, which have made the draft stronger. While the work focuses on toy models, deep understanding of small models is still valuable and arguably paves the way for understandability of larger and more complex ones.

However, due to the following concerns raised by reviewers and AC’s own read, this draft may not be ready for publication as a full paper yet. After adjusting the paper’s claims, it could still make a good workshop paper.

The main concerns stem from poor positioning of this work. The work might have originated prior to other work on the topic of “grokking” which is admirable, but to meet ICLR’s standards, it needs to correctly reflect the state of the art. Presenting known results as novel findings is one issue. But I think the more problematic issue is arguing for an interpretation that has already been disproven.
- There is experimental evidence that there are alternative generalizing algorithms that could have completely different forms, even without a periodic structure. See [1].
- On a similar note, there is some gap between theory and practice. The experimental evidence is not strong enough to show that all models trained from scratch would converge to this solution, though that seems to be the position of the paper.
- While the derivation of the exact solution is interesting and slightly different from prior work, the high-level contributions of this draft are limited. The authors claim that the main contributions of the paper are:  *“It addresses two main questions. (i) What is necessary and not necessary for grokking to occur? It turns out that weight decay, CSE loss, sharp weight norm spikes, gradient noise, flatter minima, or transformer architecture are not necessary. Any form of non-linearity and capacity for feature learning are necessary. My work is the first to narrow it down to a very simple model. (ii) How universal is the solution found by optimization? The universality is demonstrated by comparing empirically obtained networks to the analytic solution.”* However, all elements of (i) are already present in prior work, some of which are already cited in this paper, without properly discussing their findings. And there is clear evidence against (ii) as demonstrated in [1].

I find some of the author responses incorrect, or it appears that they are unwilling to take into account reviewer feedback. For example:
- *“At some point in training (and assuming there is enough data), the only way to decrease the training loss is to start learning the right features.”* is incorrect (one can easily find counter examples).
- The response to considering a true OOD test set is not satisfying.
- The dismissal of any continuous progress measure in general seems too strong given their limited experimental evidence on weight norms.
- The discussion of $a_c$ seems overly limiting, and there is empirical evidence against *“The critical amount of data $a_c$ depends strongly on the modular operation we are trying to learn and weakly on the optimizer."* For example, see [3] for the interaction between different hyperparameters, in particular dataset size, regularization strength, and model size. It provides experimental evidence for the connection between optimizer’s weight decay parameter and the minimal amount of data leading to observing “grokking”.
- *“Pure attention architecture was not studied in the context of modular arithmetic or other algorithmic datasets, to the best of my knowledge.”*  See [1].

Overall, this paper covers an interesting and important topic in mechanistic interpretability and has lots of potential. However, it may not be ready for publication.

1. https://arxiv.org/pdf/2306.17844.pdf
2. https://arxiv.org/pdf/2210.01117.pdf
3. https://pair.withgoogle.com/explorables/grokking/

**Justification For Why Not Higher Score:**

The author's responses seem to violate double-blind policy. Regardless of that, I believe the draft itself suffers from several issues as also discussed in the meta review:

The main concerns stem from poor positioning of this work. The work might have originated prior to other work on the topic of “grokking” which is admirable, but to meet ICLR’s standards, it needs to correctly reflect the state of the art. Presenting known results as novel findings is one issue. But I think the more problematic issue is arguing for an interpretation that has already been disproven.
- There is experimental evidence that there are alternative generalizing algorithms that could have completely different forms, even without a periodic structure. See [1].
- On a similar note, there is some gap between theory and practice. The experimental evidence is not strong enough to show that all models trained from scratch would converge to this solution, though that seems to be the position of the paper.
- While the derivation of the exact solution is interesting and slightly different from prior work, the high-level contributions of this draft are limited. The authors claim that the main contributions of the paper are:  *“It addresses two main questions. (i) What is necessary and not necessary for grokking to occur? It turns out that weight decay, CSE loss, sharp weight norm spikes, gradient noise, flatter minima, or transformer architecture are not necessary. Any form of non-linearity and capacity for feature learning are necessary. My work is the first to narrow it down to a very simple model. (ii) How universal is the solution found by optimization? The universality is demonstrated by comparing empirically obtained networks to the analytic solution.”* However, all elements of (i) are already present in prior work, some of which are already cited in this paper, without properly discussing their findings. And there is clear evidence against (ii) as demonstrated in [1].

I find some of the author responses incorrect, or it appears that they are unwilling to take into account reviewer feedback. For example:
- *“At some point in training (and assuming there is enough data), the only way to decrease the training loss is to start learning the right features.”* is incorrect (one can easily find counter examples).
- The response to considering a true OOD test set is not satisfying.
- The dismissal of any continuous progress measure in general seems too strong given their limited experimental evidence on weight norms.
- The discussion of $a_c$ seems overly limiting, and there is empirical evidence against *“The critical amount of data $a_c$ depends strongly on the modular operation we are trying to learn and weakly on the optimizer."* For example, see [3] for the interaction between different hyperparameters, in particular dataset size, regularization strength, and model size. It provides experimental evidence for the connection between optimizer’s weight decay parameter and the minimal amount of data leading to observing “grokking”.
- *“Pure attention architecture was not studied in the context of modular arithmetic or other algorithmic datasets, to the best of my knowledge.”*  See [1].

Overall, this paper covers an interesting and important topic in mechanistic interpretability and has lots of potential. However, it may not be ready for publication.

1. https://arxiv.org/pdf/2306.17844.pdf
2. https://arxiv.org/pdf/2210.01117.pdf
3. https://pair.withgoogle.com/explorables/grokking/

**Justification For Why Not Lower Score:**

N/A

---

### Decision · Program_Chairs · 2024-01-16

Reject